# Discovering molecular features of intrinsically disordered regions by using evolution for contrastive learning

**Alex X. Lu**[1¤a], **Amy X. Lu**[1¤b], **Iva Pritišanac**[2,3¤c], **Taraneh Zarin**[2¤d], **Julie D. Forman-Kay**[3,4], **Alan M. Moses**[1,2]*

**1** Department of Computer Science, University of Toronto, Toronto, Canada, **2** Department of Cell and Systems Biology, University of Toronto, Toronto, Canada, **3** Program in Molecular Medicine, Hospital for Sick Children, Toronto, Canada, **4** Department of Biochemistry, University of Toronto, Toronto, Canada

¤a Current address: Microsoft Research, Cambridge, Massachusetts, United States of America
¤b Current address: Department of Electrical Engineering and Computer Sciences, Berkeley, California, United States of America
¤c Current address: Molecular Biology and Biochemistry Medical University of Graz, Graz, Austria
¤d Current address: Systems Biology Program, Center for Genomic Regulation, Barcelona, Spain
* alan.moses@utoronto.ca

**Data Availability Statement:** Code for training our models and visualizing/extracting features is available under a CC-BY license at github.com/alexxijielu/reverse_homology/. Pretrained weights

## Abstract

A major challenge to the characterization of intrinsically disordered regions (IDRs), which are widespread in the proteome, but relatively poorly understood, is the identification of molecular features that mediate functions of these regions, such as short motifs, amino acid repeats and physicochemical properties. Here, we introduce a proteome-scale feature discovery approach for IDRs. Our approach, which we call "reverse homology", exploits the principle that important functional features are conserved over evolution. We use this as a contrastive learning signal for deep learning: given a set of homologous IDRs, the neural network has to correctly choose a held-out homolog from another set of IDRs sampled randomly from the proteome. We pair reverse homology with a simple architecture and standard interpretation techniques, and show that the network learns conserved features of IDRs that can be interpreted as motifs, repeats, or bulk features like charge or amino acid propensities. We also show that our model can be used to produce visualizations of what residues and regions are most important to IDR function, generating hypotheses for uncharacterized IDRs. Our results suggest that feature discovery using unsupervised neural networks is a promising avenue to gain systematic insight into poorly understood protein sequences.

## Author summary

Intrinsically disordered regions (IDRs) are widespread in proteins but are poorly understood on a systematic level because they evolve too rapidly for classic bioinformatics methods to be effective. We designed a neural network that learns what features (for example, electrostatic charge, or the presence of certain motifs) might be important to the function

for our models, fasta files of IDR sequences used to train both models, and labels for IDRs used in our classification benchmarks are available at zenodo.org/record/5146063.

**Funding:** This research was supported by NSERC Discovery Grant RGPIN-2018-04924, CIHR Project Scheme PJT-148532 to AMM and Canada Research Chairs to AMM and JDFK. This work was funded by an NSERC-CGS grant to AXL. This work was partially performed on a GPU generously donated by NVIDIA to AMM. The funders had no role in study design, data collection and analysis, decision to publish, or preparation of the manuscript.

**Competing interests:** I have read the journal's policy and the authors of this manuscript have the following competing interests: AMM is a Consultant to Dewpoint Therapeutics Inc.

of IDRs, even when we don't have prior knowledge of function. Our neural network learns by exploiting principles of evolution. Important features tend to be conserved over species, so guessing what sequences evolved from the same common ancestor helps the neural network identify these features. Importantly, training a neural network this way can be defined as a fully automatic operation, so no manual effort is required. After our neural network is trained, we can apply interpretation techniques to understand what kinds of features are important to IDRs globally in the proteome, and to form hypotheses about specific IDRs. We show that many of the features our neural network learns are consistent with features we already know to be important to IDRs. We hope that our neural network can be applied to help biologists form hypotheses about poorly characterized IDRs.

## Introduction

Despite their critical role in protein function, the systematic characterization of intrinsically disordered regions (IDRs) remains elusive [1–3]. IDRs comprise of about 40% of the residues in eukaryotic proteomes [4]. Unlike structured domains, IDRs do not fold into a stable secondary or tertiary structure, and this lack of structure helps facilitate many key functions. For example, some IDRs mediate protein-protein interactions, because their lack of structure allows them to adapt their conformation to different interaction partners [4,5].

Databases of large-scale predicted [6] and experimentally confirmed [7] intrinsically disordered regions are available. Prediction of intrinsically disordered regions from primary amino acid sequences is a well-developed area of research in computational biology [8]. On the other hand, aside from defining a peptide region as intrinsically disordered, relatively little can be predicted about its function [2,3], although there has been considerable effort to identify binding sites [9,10], conditionally folded regions [11] and more recently, to predict disordered region functions [12,13]. This is in contrast to the situation for folded protein regions, where highly specific predictions of function based on sequence using such universal resources as BLAST [14] and Pfam [15].

Other approaches have been devised to classify intrinsically disordered regions into functional groups, initially into a small number of groups based on predicted biophysical properties [16,17]. Using larger numbers of molecular features (including biophysical properties, matches to short motifs and repeats, and residue composition) and their patterns of evolution [18] we showed strong association between ~20 biological functions and the molecular features of intrinsically disordered regions. Indeed, conserved molecular features were used as input for general predictions of IDR functions [13].

The features important to function of disordered regions are highly diverse. The best understood features are "short linear motifs", peptides of 4–12 residues [19]. In some cases, multiple copies or local clustering of motifs is necessary for function [20]. Other IDRs depend upon global "bulk" features that are distributed through the entire sequence. For example, mitochondrial import IDRs require the sequence to be positively charged and hydrophobic [21], certain phase-separating proteins require IDRs with many R/G repeats that facilitate condensate-forming interactions [22], and alternating positive and negative charged regions in an IDR of the cell-cycle regulating protein p27 mediates the strength of phosphorylation of key regulatory sites [23].

Our knowledge of features important to IDRs is not likely to be comprehensive. Indeed, features are continuously being discovered as research on IDRs develops: recently characterized features include aromatic amino acid patterning for prion-like domains [24,25] or

hydrophobic residues for activation domains [26,27]. We therefore set out to design a systematic computational method for discovering features in IDRs, that is unbiased by prior knowledge or interest. The problem of feature discovery runs closely parallel to the concept of motif discovery [28,29]: given a set of functionally related sequences, motif discovery methods attempt to find overrepresented subsequences with the idea that these motifs may represent conserved binding, interaction, or regulatory sites informative of the function of proteins. Motif discovery approaches range from fully unsupervised to regression approaches where function is predicted from sequence. Among the most successful strategies for motif discovery are those that exploit the principle that important functional motifs are conserved over evolution [30–32]. Because comparative sequence data is available at genomic and proteomic scales, and is unbiased by a particular experimental condition or research question, comparative genomic and proteomic approaches have the potential to discover large numbers of functional motifs. However, alignment-based approaches to find conserved motifs in IDRs identify only a small minority (~5%) of the residues in IDRs [33]; short motifs of about 2–10 residues often occur as small islands of conservation in IDRs that have no detectable sequence similarity otherwise [34]. These short conserved elements are not expected to describe the "bulk" molecular properties such as charge, hydrophobicity or motif density, that are expected to be important for IDR function and appear to be conserved during evolution [35–37].

In order to develop a proteome-scale feature discovery approach capable of using evolution to learn more expressive features than motifs, we applied neural networks. To learn biologically relevant features, neural networks must be asked to solve a training task (i.e. a pre-specified loss function) [38]. Approaches to infer sequence function using neural networks employ regression tasks, where models learn to predict high-throughput measurements [39–43]. For example, training genomic sequence models on labels representing the presence or absence of transcription factor binding leads to the model learning features that directly correspond with the consensus motifs for these transcription factors [44]. Similarly, training neural networks to predict high-throughput measurements [45,46] of activation domain function highlighted clusters of hydrophobic residues within acidic regions [26,27] as a key sequence feature. While these supervised approaches discover important features, we reasoned that they would only learn features relevant to the specific training task.

Instead, we sought to use evolutionary conservation as a learning signal. Since orthologous sequences can be automatically obtained using sequence comparison and gene order [47,48], labels about homology can be automatically obtained for IDRs. We therefore investigated self-supervised learning. Self-supervised learning trains models on "proxy" tasks resembling play and exploration [49], for which the labels can be automatically generated from data. These tasks are not directly useful, but are intended to teach the model transferable skills and representations, and are designed so the models learn autonomously without expert labels. Several self-supervised learning approaches have been applied to protein sequences, and have been effective in teaching the models features that are useful for downstream analyses [50–55]. However, the majority of these directly repurpose tasks from natural language processing [50–53], and it is unclear what kinds of features the tasks induce the models to learn in the context of protein sequences.

We designed a new self-supervised method that purposes principles in comparative genomics as a learning signal for our models. While IDRs generally cannot be aligned over long evolutionary distances [56], they can still be considered homologous if they occur at similar positions in homologous proteins [18]. Given a subset of homologous IDRs, our model is asked to pick out a held-out homologue, from a large set of non-homologous sequences. This task, which we call reverse homology, requires our model to learn conserved features of IDRs, in order to distinguish them from non-homologous background sequences. Our method is a

contrastive learning method, a strategy that is now frequently employed in self-supervised learning [54,57–60]. We show that reverse homology can be applied on a proteome-wide scale to learn a large set of diverse features. While these "reverse homology features" are learned by the neural network to solve the reverse homology proxy task, we show that they can be visualized and interpreted, are enriched in functional terms, and can be purposed for bioinformatics analyses that yield hypotheses connecting specific features to function. Taken together, our work demonstrates that deep learning methods can be designed to identify sequence features that are preserved over evolution (without relying on sequence alignment), and that feature discovery for biological sequences is an unexplored application of self-supervised learning.

## Results

### Reverse homology

To learn functional features of IDRs unbiased by prior knowledge, we propose a novel self-supervised proxy task that uses evolutionary homology between protein sequences to pose a contrastive learning problem. Homologous proteins derive from a shared evolutionary ancestor, and will frequently share similar functions [61]. For full proteins and structured domains, homology can be reliably identified based on sequence similarity [61]. IDRs are usually flanked by structured regions that are easily identified as homologous [36]. Since these structured regions will usually align well, and since the order of domains is usually strongly conserved in proteins [62], IDRs that occur at the same position across homologous proteins in a multiple-sequence alignment can be considered to be homologous even when they share little sequence similarity (Fig 1A) [63–66]. As bioinformatics tools can accurately annotate what parts of a protein are IDRs [8], defining homologous groups of IDRs using multiple-sequence alignments across the entire proteome can be defined as a fully automated operation [18].

We will use these sets of homologous IDRs (Fig 1B) as the basis for our proxy task (Fig 1C, see Methods for a more detailed definition.) Given a set of homologous IDRs, a neural network (Fig 1D) is asked to determine which sequence is a held-out homologue from a set of IDRs where the other sequences are randomly drawn non-homologous sequences (Fig 1E). We call this task "reverse homology", because it "reverses" the typical sequence homology search process, where we have a target sequence of unknown homology, and we search across many query sets or sequences to assign homology [61]. In our task, we give the model a query sequence of known homology, and ask it to determine if target sequences are homologous or not.

In previous work, we explained the theoretical principles behind using evolutionary homology as a basis for contrastive learning [57]: our method is expected to learn conserved features of protein sequences, which we argue are likely important for the conserved function of rapidly-diverging IDRs (S1 Methods).

We use both max and average pooling in our convolutional neural network architecture (Fig 1D) to reflect different ways in which local features can contribute to functions of IDRs. We reasoned max pooling, which identifies a single window that maximally activates (i.e. creates the highest feature value for) the feature, would capture function determined by presence or absence; for example, a single SH3 binding motif [67] may be sufficient for recognition and function. On the other hand, other functions require multiple copies of a feature [20], a certain proportion of the sequence to have a feature [22], or scale as more of the feature is present [36]. We reasoned average pooling, which produces the average activation value across all windows, would allow for the capture of additive distributed properties within the receptive field of the convolutional layers. This architecture facilitates interpretation (see Methods): we pair our trained models with several neural network interpretation methods to understand the

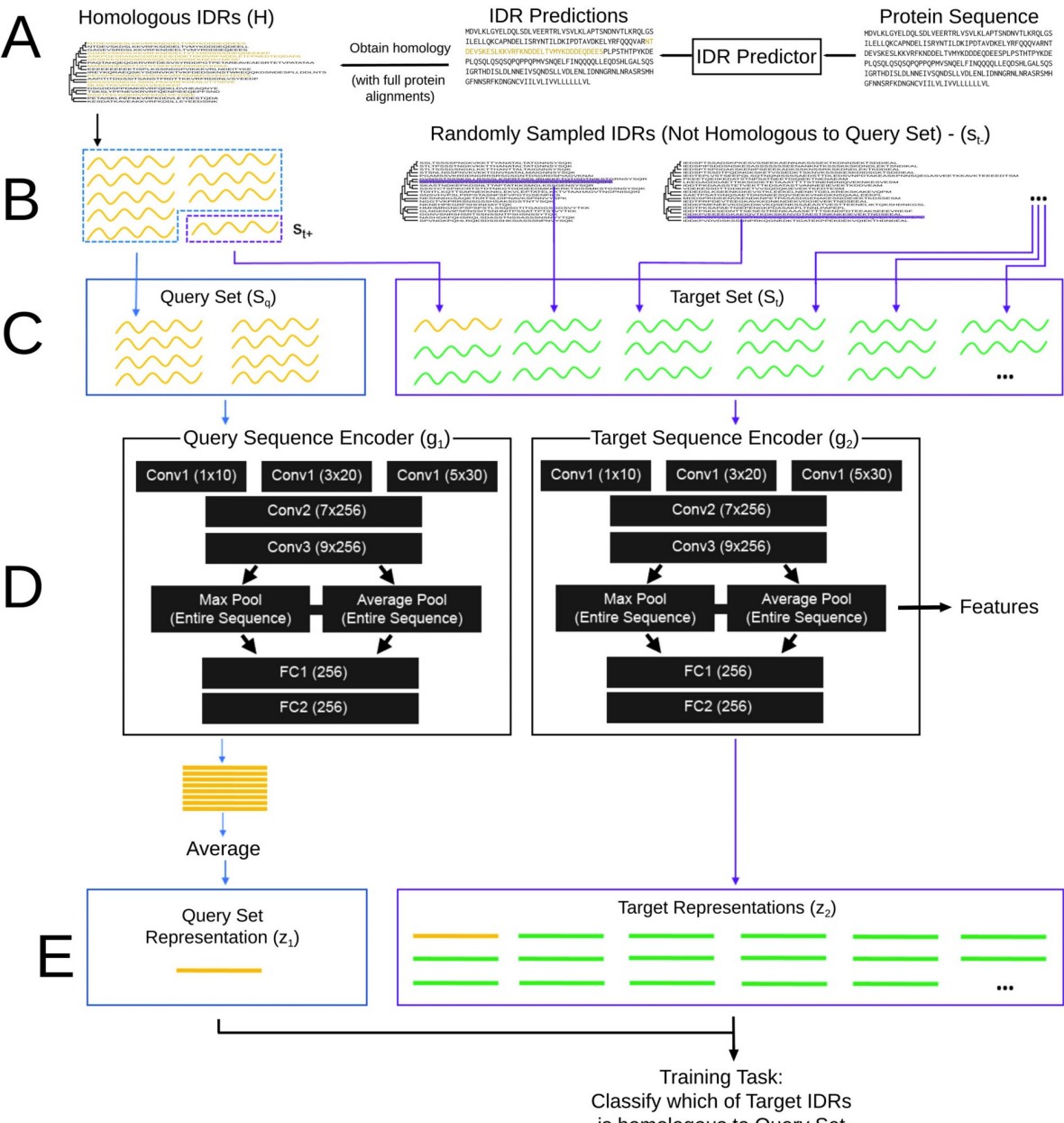

**Fig 1. A schematic description of the reverse homology method.** A) We use standard intrinsically disordered region (IDR) prediction methods to obtain predicted IDRs for the whole proteome. We then extract homologous sets of disordered regions from whole protein multiple sequence alignments of orthologs, obtained from public databases B) Homologous sets of IDRs (gold) are combined with randomly chosen non-homologous IDRs to derive the proxy task for each region C) We sample a subset of IDRs (blue dotted box) from $H$ and use this to construct the query set ($S_q$, blue box). We also sample a single IDR (purple dotted box) from $H$ not used in the query set and add this to the target set ($S_t$, purple box). Finally, we populate the target set with non-homologous IDRs (green), sampled at random from other IDRs from other proteins in the proteome. D) The query set is encoded by the query set encoder $g_1$. The target set is encoded by the target set encoder $g_2$. In our implementation, we use a five-layer convolutional neural network architecture. Both encoders include both max and average pooling of the same features, which correspond to motif-like and repeat or bulk features, respectively. We label convolutional layers with the number of kernels x the number of filters in each layer. Fully connected layers are labeled with the number of filters. E) The output of $g_1$ is a single representation for the entire query set. In our implementation, we pool the sequences in the query set using a simple average of their representations. The output of $g_2$ is a representation for each sequence in the target set. The training goal of reverse homology is to learn encoders $g_1$ and $g_2$ that produce a large score between the query set representation and the homologous target representation, but not non-homologous targets. In our implementation, this is the dot product: $g_1(S_q) \cdot g_2(s_{t+}) > g_1(S_q) \cdot g_2(s_-)$. After training, we extract features using the target sequence encoder. For this work, we extract the pooled features of the final convolutional layer, as shown by the arrow in D.

features learned. In principle, this architecture can be replaced with many others, but we leave exploration of possible architectures to future work (see Discussion).

## Reverse homology learns a diverse range of features for yeast intrinsically disordered regions

We first trained a reverse homology model using 5,306 yeast IDR homology sets containing a total of 94,106 sequences. We focused on the features from the final convolutional layer of the target encoder, as our goal is to identify interpretable features in individual intrinsically disordered regions. To obtain a global view of the molecular feature space learned by the neural networks, we produced a UMAP scatterplot [68], where each point represents a feature, using the correlation distance between activation values across IDR sequences for each feature (Fig 2). We generated sequence logos for each feature by adapting a recently proposed approach for DNA convolutional features [44] (see Methods). Note that the width of the average pool sequence logos are arbitrarily chosen to be the same width of the max pool sequence logo (which corresponds to the receptive field of the convolutional layers) even though these features pool information across the entire sequence. Examination of the most activating max-pool and average-pool IDR for each feature highlights the differences in the types of sequences these features are recognizing (S1 File).

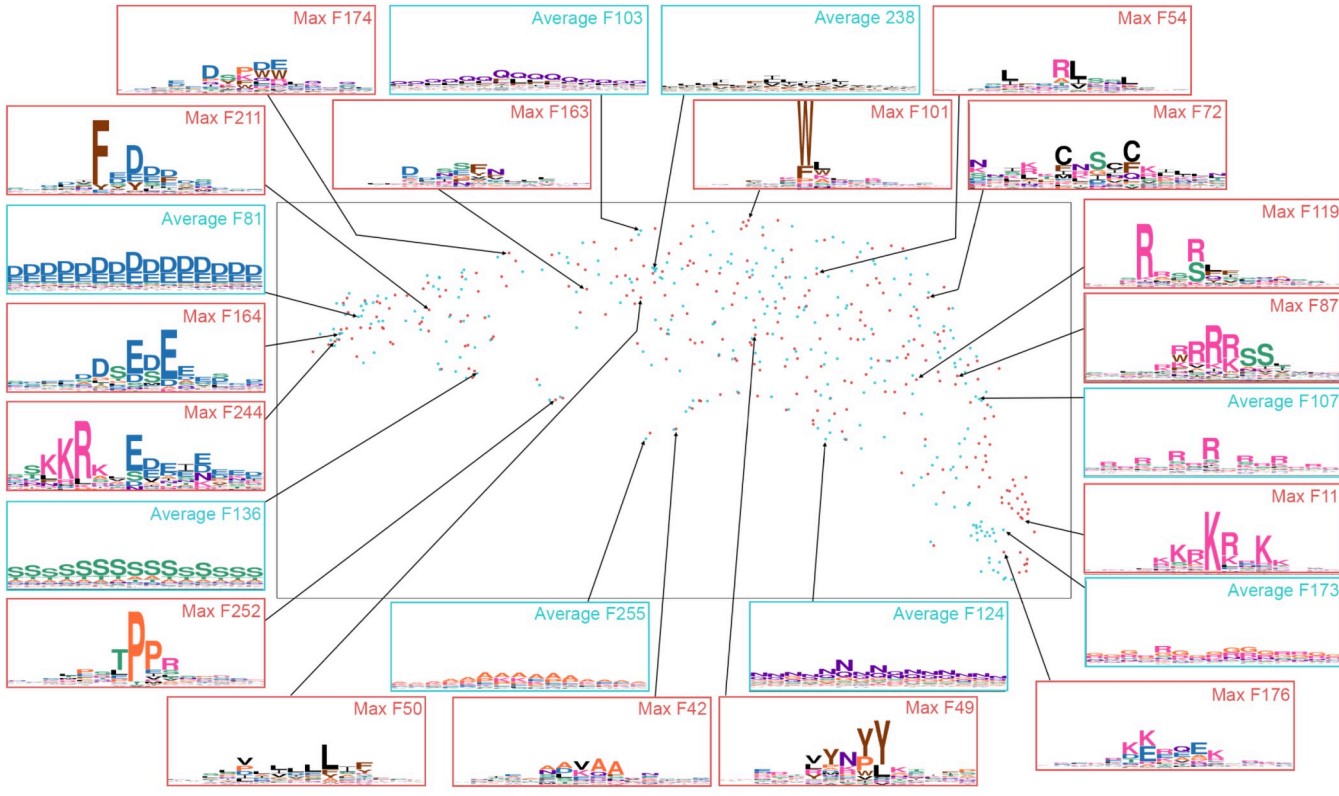

**Fig 2. UMAP scatterplot of reverse homology features for our yeast model.** Reverse homology features are extracted using the final convolutional layer of the target encoder: max-pooled features are shown in red, while average-pooled features are shown in blue. We show the sequence logo corresponding to select features, named using the index at which they occur in our architecture (see Methods for how these are generated). Amino acids are colored according to their property, as shown by the legend at the bottom. All sequence logos range from 0 to 4.0 bits on the y-axis.

Overall, we observed four major axes of features. To the left of the scatterplot, we observed features with negatively charged amino acids (e.g., Average F81 and Max F164). Features with positively charged amino acids were concentrated at the bottom right (e.g. Max F11 and Average F173). Features containing hydrophobic amino acids are at the top of the distribution (e.g., Average F238, Max F54, and Max F72). Finally, we observed features rich in uncharged polar amino acids (e.g., Average F136, Max F252 and Average F124) or alanine (e.g., Average F255 and Max F42) scattered along the bottom of our UMAP. Features in between these poles often exhibited a mixture of properties. For example, Max F211, Max F174, and Max F163 all contain both negative and hydrophobic residues, Max F54 contains both positive and hydrophobic residues, and Max F49 contains both aromatic and polar amino acids.

Specific features captured both motifs and bulk properties known to be important for IDR function. As examples of motifs, Max F252 is consistent with the TPP phosphorylation motif [69], while Max F87 is similar to the PKA phosphorylation motif RRxS [70]. As examples of bulk properties, Average F173 captures RG repeats important for phase separation [22], while other average features look for combinations of amino acids with similar biochemical properties (Average F136 measures S/T content, Average F124 measures N/Q content, and Average F81 measures acidic amino acids D/E). Finally, other features captured patterns that we were not able to associate with previously known IDR properties: for example, Average F107 captures spaced out arginine repeats (e.g. RxRx), and Max F244 captures a window of positive to negative charge transition. We hypothesize these features could represent charge patterning in IDRs [71,72].

Overall, we were able to identify 70 of 512 features as complete or partial matches to motifs or bulk features previously considered to be important to IDRs (S2 File). We consider this number a lower bound as there are features that we were uncertain about. Together, our global analysis demonstrates that reverse homology induces our model to learn a wide diversity of biochemically sensible features.

## Reverse homology features are predictive of yeast IDR function and correlated with previous literature-curated features

Having qualitatively confirmed that our model learns diverse features, our next goal was to more quantitatively evaluate them. To do this, we first we compared reverse homology features to literature-curated features by Zarin *et al.* [18]. We reasoned that a good measure of consistency between features detected by our neural network and literature-curated features is having the neural network feature activated at the same positions in amino acid sequences as the literature-curated feature. For each of these features that can be expressed as regular expressions (37 motifs, 7 physicochemical properties, 8 amino acid frequencies and 14 repeats) we computed the correlation of the activation for each of the features of our trained reverse homology model, and selected the feature with the maximum correlation (see Methods). Because many of the literature-curated features are relatively simple and reflect only single amino acid repeats, or relationships between subsets of 2–3 amino acids, they may be easy to capture by chance given the large number of features (512). Therefore, to set the expectation for these maximum correlations, we compared the features in the trained model to a randomly initialized model, as random untrained models have shown to be a strong baseline for protein representation learning problems [73,74]. (Fig 3A, please see S3 File for a table of all features, their regular expressions, and the maximal correlations with our trained and random models).

In line with the idea that the reverse homology trained neural network encodes many features that are similar to previously known features (S2 File), we observe higher correlations with 55 of 66 literature-curated features with our reverse homology model than an untrained

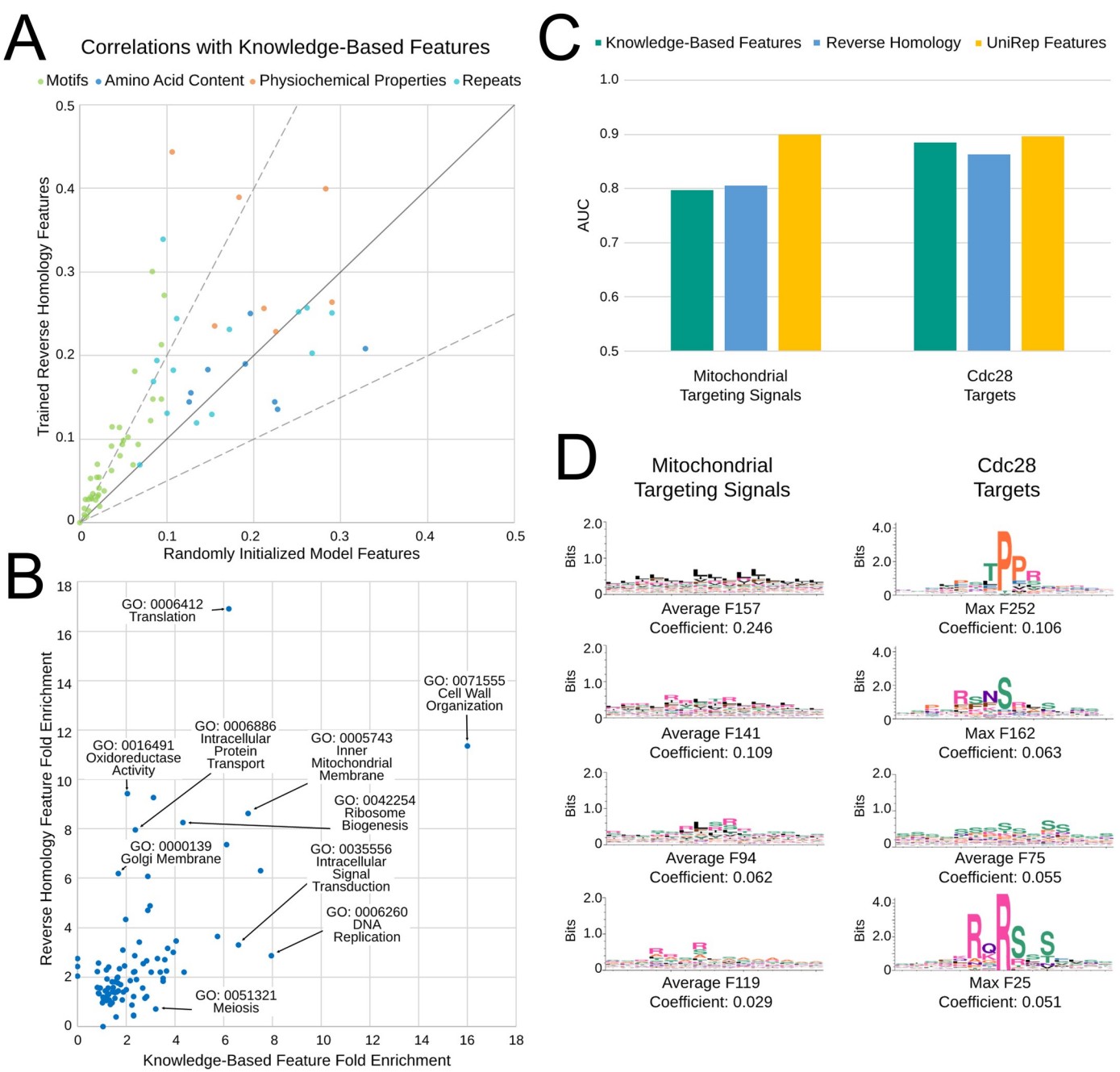

**Fig 3.** A) The maximum correlation between features in the final convolutional layer and each of the 66 literature-curated features from the trained reverse homology model vs. a randomly initialized model. Features are coloured by their category (top legend). Black trace indicates y = x, while grey traces indicate features more than 2.0x correlated, and less than 0.5x times correlated than the untrained random features. B) Fold enrichment for the set of nearest neighbors using feature representations from the final convolutional layer of the target encoder of our reverse homology model, versus literature-curated feature representations, for 92 GO Slim terms. We show the names of some GO terms in text boxes. C) Area under the receiver operating curve (AUC) for regularized logistic regression classification of mitochondrial targeting signals and Cdc28 targets obtained through 5-fold cross validation. A deep language model (Unirep, gold) performs better than reverse homology (blue) and literature-curated features (green). D) Features with largest coefficients (indicated below each logo) selected by the sparse classifier are consistent with the known amino acid composition biases in mitochondrial targeting signals (left) and short linear motifs in Cdc28 substrates (right).

random model. Features corresponding to motifs are learned significantly better by our reverse homology model than the random model (n = 37, paired t-test p-value 4.63E-06; mean 0.042 and standard deviation 0.0475). Features corresponding to physicochemical properties (n = 7; p-value 0.067; mean 0.1091 and standard deviation 0.1266) and repeats (n = 14; p-value 0.077; mean 0.042 and standard deviation 0.082) are more correlated, but with less confidence than motifs. In contrast, features learned by our model are not more correlated with features capturing single amino acid content compared to a randomly initialized model, and are in fact, slightly less correlated on average (n = 8; p-value 0.4409; mean -0.02 and standard deviation 0.067). These results are consistent with the intuition that shorter repeats of one or two amino acids are more likely to be present in features by random chance, but longer and more specific combinations like motifs must be learned.

We next compared the enrichments for GO terms (Table C in S4 File) of reverse homology features to literature-curated features (Fig 3B). To compute enrichment for each feature, we counted proteins that had a protein with the same GO term as their nearest neighbor in the reverse homology feature space, and compared the fraction to the background. Importantly, this analysis includes all GO Slim terms (curated by the SGD database [75]) with at least 50 proteins in our set of proteins with IDRs (for a total of 92 terms), so it is not biased towards functions previously known to be associated with IDRs.

Overall, we find that while some GO Slim categories are highly enriched with both feature sets (e.g. "Cell wall organization", "Translation" or "Ribosome biogenesis", highlighted in Fig 3B), other categories are much more enriched with our reverse homology features than literature-curated features (e.g. "Oxidoreductase Activity", "Intracellular Protein Transport", or "Golgi Membrane"). Conversely, some categories are more enriched using literature-curated features (e.g. "Meiosis", "Intracellular Signal Transduction", or "DNA replication"). These results suggest that the neural network is learning features relevant to biological processes that are different from the ones associated with literature-curated features.

Next, we benchmarked vector representations of IDRs extracted using our model compared to self-supervised protein representations based on language models trained on large unlabelled protein databases [50–52] and knowledge-based features curated from literature [18] (Tables A and B in S4 File). We compared these representations on a series of classification problems predicting various aspects of IDR function (Tables A, B and C in S4 File; details on classification datasets, classifiers, and baselines are also in S4 File). Overall, we observe that the language models performed best. As expected for IDR sequences, sequence-similarity (measured using blastp E-values [14]) is a poor predictor of function in these benchmarks. For our reverse homology representation, we observed a trade-off depending on layer between the performance of the representation on our benchmark tasks and interpretability. Representations from the final fully connected layer of our target encoder perform comparably to other self-supervised protein representation learning methods, suggesting that our model can represent IDRs at a similar level of performance as the language models, but with a lower-parameter architecture and substantially less training data. However, the features in this layer are less interpretable than the convolutional layers. Representations from our final convolutional layer still outperform the literature-curated features at most problems, suggesting that these features may still encode more functional information than expert-curated features.

For example, we compared the predictive power (Fig 3C) of these representations to predict two highly-specific IDR functions in yeast, mitochondrial targeting signals and direct Cdc28 phosphorylation [13]. On both tasks we find that Unirep [52] performs best (Table B in S4 File) achieving 5-fold cross-validation AUC of 0.9 on both tasks. On the other hand, we also used a sparse regression model (trained on the entire dataset) to select the most predictive reverse homology features and found that they are readily interpretable (Fig 3D). The top

features for mitochondrial targeting signals are average pool features rich in hydrophobic and positively charged residues, as expected based on the known residue composition biases of these signals [21]. In contrast, the features for Cdc28 targets are mostly motifs, with the top ranked feature matching the specificity of a proline-directed kinase (such as Cdc28) and the average pool feature is rich in serines and prolines, consistent with known multi-site phosphorylation in these substrates [20]. Thus, predictive models based on reverse homology features appear as interpretable as those based on knowledge based features [13], in contrast to features obtained from language models [50,52].

## Features that recognize bulk properties associated with cell wall maintenance and phase separation

Having confirmed that the features learned through reverse homology are diverse and associated with known features and functions of IDRs, we next sought analyze individual reverse homology features. First, we considered average pool features recognizing S/T repeats (Average F136 –Fig 4A) and RG repeats (Average F65 –Fig 4C). These exemplify "bulk properties" of IDRs that, to our knowledge, no computational method has previously been designed to identify.

Long regions of S/T-rich segments are often sites of *O*-glycosylation in yeast proteins [76]. A previous computational analysis revealed that fungal proteins with an extremely high proportion of S/T-rich regions in their sequence are often cell wall proteins involved in maintenance of the cell wall [76]. Consistent with this, we find an enrichment (using the GOrilla tool [77]) for cell wall proteins (15/31, q-value 3.16E-16), cell wall organization or biogenesis proteins (20/31, q-value 1.69E-15), and extracellular region proteins (17/31, q-value 4.15E-20) in the proteins with IDRs that highly activate Average Feature 136 an S/T repeat feature (Fig 4A, left). We observed that the top 3 IDRs are all cell wall proteins: Cwp1 [78] and Tir3 [79] are cell wall mannoproteins, while Wsc2 is involved in maintenance of the cell wall under heat shock [80]. Our 4[th] ranked IDR is in Uth1, which is predominantly known as a mitochondrial inner membrane protein [81]. However, deletion of Uth1 alters the polysaccharide composition of the cell wall, with mutants being more robust to lysis conditions, leading to the argument that Uth1's role at the cell wall, not the mitochondria, better explains its functions in cell death [82].

To visualize the activation of Average Feature 136 IDR in Uth1 in closer detail, we produce mutation maps (adapted from [39], also referred to as *in silico* or computational mutagenesis, e.g., [44]) (Fig 4B). We substituted each amino acid position in the IDR to every other amino acid, and measured the change the mutation induces in the value of Average Feature 136. We visualize these mutation maps as "letter maps" (not to be confused with sequence logos, which represent the variability of columns in multiple sequence alignments [83]) shown in Fig 3 (see Methods for details). In these letter maps, residues that the feature favors (i.e., would generally result in a drop in the feature if mutated) are shown above the axis, while residues the feature disfavors are shown below the axis. For favored positions, we show the amino acids that are most favored, and these may not always be the wild-type amino acid, as the feature can favor the wild-type, but favor other amino acids more. For disfavored positions, we show the amino acids that are more disfavored (see Methods). Overall, analyzing the IDR at 52–104 in Uth1 reveals long tracts of S, T, and A-rich regions that are favored by Average Feature 136 (Fig 4B).

Similarly, RG repeats are found in phase-separating RNA-binding proteins that form membraneless organelles [22]. Consistent with this, we found an enrichment for RNA-binding (13/20, FDR q-value 6.37E-5) proteins and for proteins localizing to the ribonucleoprotein complex (9/20, q-value 1.47E-1) in the IDRs that most strongly activate Average Feature 65.

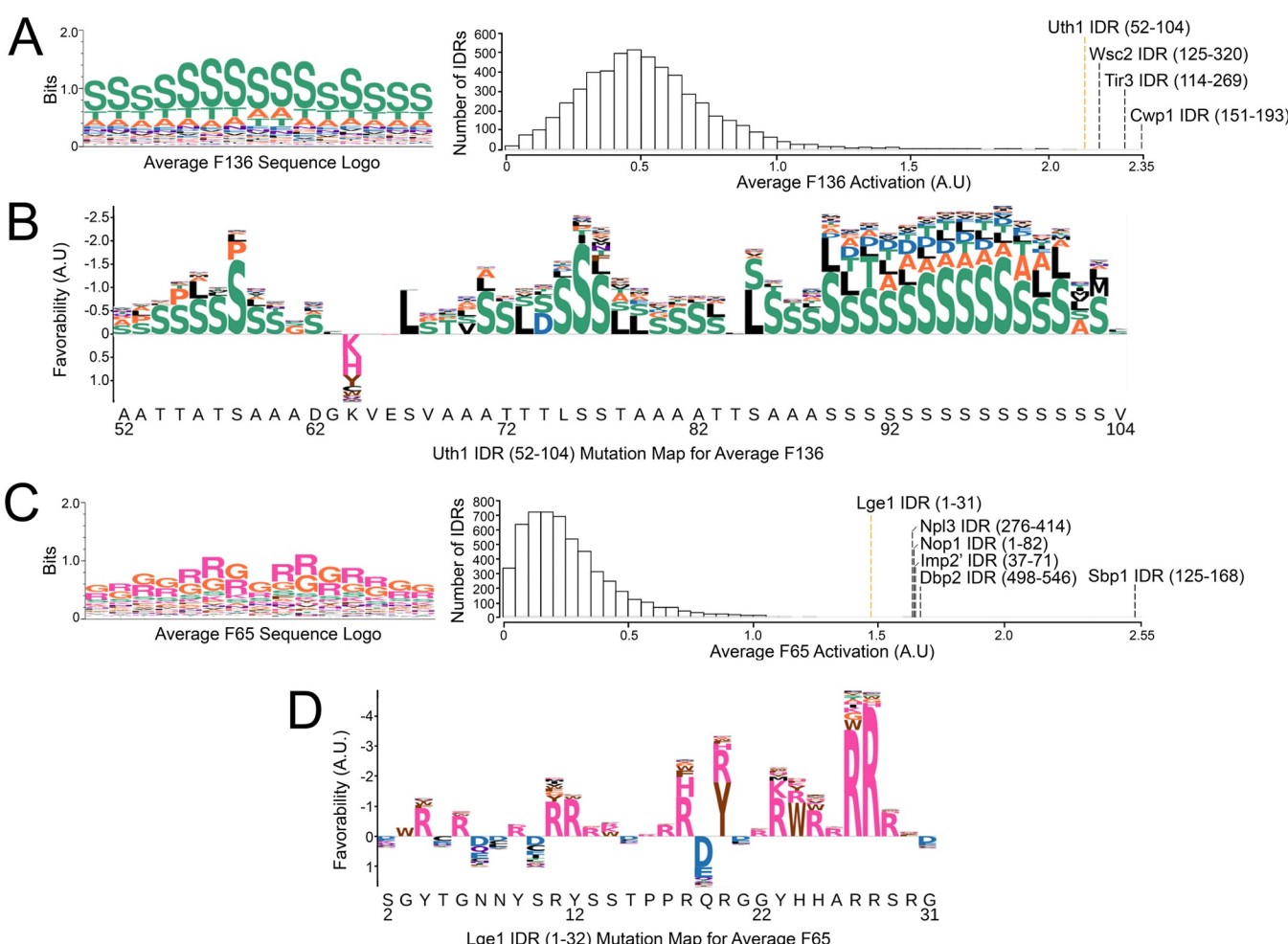

**Fig 4. Sequence logos, feature distributions, and examples of mutation maps for each average feature.** (A,C) Sequence logos and a histogram of the value of the feature across all IDRs is shown for Average F136 (A) and Average F65 (C). We annotate the histograms with the top activating sequences. (B,D) Mutation maps for F136 for an IDR in Uth1 in B and for F65 for an IDR in Lge1 (D), which are the 4[th] and 6[th] most activating sequences for their respective features. Mutation maps are visualized as letter maps, where positions above the axis are positions where retaining the original amino acid is preferable, while positions below the axis are positions where the activation could be improved by mutating to another amino acid. The height of the combined letters corresponds to the total magnitude of the change in the feature for all possible mutations (which we define as the favourability). For positions above the axis, we show amino acids that result in the highest value for the feature (i.e. the most favored amino acids at that position.) For positions below the axis, we show amino acids that result in the lowest value for the feature (i.e. the most disfavored amino acids at that position).

Indeed, for our RG-repeat, Average Feature 65, 4 out of 6 of the top IDRs are proteins with known stretches of RG-repeats (Dbp2 [84]), and 3 are phase-separating proteins mediated by interactions between RG-rich regions (Sbp1 [85], Npl3 and Nop1 [22]). Interestingly, while Lge1 is also known to phase-separate through its N-terminal IDR, also identified in our analysis, this IDR is not canonically considered an RG-rich IDR and instead has been described as an R/Y-rich region [86]. Closer analysis of Average Feature 65 applied on the Lge1 N-terminal IDR (Fig 4D) indicates that the feature also prefers Ys and other aromatic acids in addition to R; although replacing G with most other amino acids reduces the value of the feature, replacing G with R, Y, or W improves the value of the feature in most spots. The preference for aromatic amino acids in addition to RG-repeats is consistent with aromatic amino acids mediating similar pi interactions and acting as "stickers" to drive phase separation [24,87]. We hypothesize that our feature may reflect this relationship and may be recognizing

the synergy between the two types of features previously thought of as distinct (RG-repeats and R/Y-rich regions).

## Applying reverse homology to predicted human IDRs or experimentally confirmed IDRs also yields diverse features correlated with literature-curated features

Having confirmed that our reverse homology task was effective at learning features that could be interpreted to drive hypotheses about yeast biology, our next goal was to train a model for human IDRs. We trained a reverse homology model using 15,996 human IDR homolog sets for a total of 634,406 sequences (see Methods).

We qualitatively confirmed that this model was learning a similar diversity of features as our yeast model (S5 File). We were able to identify features for both short linear motifs, such as consensus motifs for phosphorylation sites or metal ion binding motifs, and bulk features like repeats or charge. As with our yeast model, we found that our features were significantly more correlated with literature-curated features than a random model (paired t-test p-value 1.589E-10, average 0.084, standard deviation 0.090). S6 File contains the maximum correlations with the 66 features we previously tested for the yeast reverse homology model (for human IDRs, no comprehensive set of features has yet been published).

Since our previous models were trained using predicted IDRs sequences, we next evaluated the features learned by a model trained on experimentally verified IDRs exclusively, using the Disprot database [7]. We trained a model on 53,050 sequences from 1,467 sets of homologs (see Methods). In addition to being an order of magnitude fewer sequences and homologs, this dataset contains homolog sets where the experimentally characterized reference IDRs are from 10 different species.

Even with this smaller and more heterogeneous dataset, we are still able to learn diverse features (S7 File). We validated the model using a set of 1,000 homolog sets based on predicted human IDRs in proteins unseen in Disprot (randomly selected from the human IDR dataset used to train the previous model). We observed that the performance of the classification task is ~ 30% (S4C Fig), much higher than expected by chance (0.25% accuracy). While this is lower than the model trained on predicted human IDRs (about 70% accuracy), the performance suggests that the features obtained from experimentally verified IDRs generalize to predicted IDRs. Second, we also verified that the features from this model are significantly more correlated with literature-correlated features than a randomly initialized model (paired t-test p-value 0.003, average 0.017, standard deviation 0.040; maximum correlations are in S8 File). We note that the difference between random and trained is smaller than our previous models, suggesting that the larger datasets of predicted IDRs contain biologically relevant variation unseen in the smaller set of experimentally characterized IDRs.

## Feature discovery using reverse homology

We next sought to test whether our approach could identify novel interpretable features. First, we compared the "motifs" discovered in our global feature discovery approach to a state-of-the-art motif finder, DALEL [88]. DALEL searches for the most strongly enriched motifs among sets of protein sequences that are hypothesized to share function. We therefore used a simple t-test to identify the reverse homology feature most enriched in two datasets where the individual functional residues have been defined: characterized human Grb2 targets [88]and characterized yeast PKA targets (the Biogrid database [89]). We find that in both cases the most enriched feature among the reverse homology feature corresponded to the known motif (Fig 5A). To compare quantitatively the residue level accuracy of the predictions from the

reverse homology features to the motifs identified by DALEL, we smoothed the activation of the top feature (see Methods) and compared the overlap of the most activated positions to the annotated functional residues. For Grb2 we found comparable performance to DALEL, while for PKA we find lower performance. Nevertheless, we found that our motif is predictive of PKA targets at the proteome level: 8 of 24 *bona fide* targets are among the top activating IDRs (defined as more than 70% of the value of the maximally activating IDR), reflecting an enrichment of 18.7 times compared to lesser-activating IDRs (8/139; Fisher exact test p-value 8.8E-08). Taken together, we find this performance in motif-finding impressive, considering that our approach learns motifs for the whole proteome, and they are not fit to any particular subset of proteins.

In visual exploration of the features learned by reverse homology models, we identified previously unknown features as well. We were interested Max Feature 231 (Fig 5B) that appears to recognize an RRRSS motif that we speculated might represent a basophilic phosphorylation site consensus, perhaps related to two overlapping PKA phosphorylation consensus motifs RRxS [70] [90]. To test this, we analyzed the frequency of two adjacent phosphorylated serines [89] in the 8-amino acid window around the maximally activated position of the top activating sequences and found 2.31 times more doubly phosphorylated regions (13 of 52 IDRs for RRRSSS vs. 15 of 139 IDRs for the feature matching the PKA consensus. Fisher exact test p = 0.020). Our results suggest that the doubly phosphorylated PKA consensus is a more widespread mechanism than currently appreciated, and illustrate the power of our unsupervised approach to discover unexpected and subtle biological patterns.

We were also interested in features in both the yeast and human models that appear to recognize transitions from regions of positive to negative charge. To our knowledge, these have not been reported, although they do resemble features that measure the patterning of positive and negative charged residues [71,72], which is generally known to be important to IDRs; our feature identifies a single local window instead of measuring these charge transitions as a property across the entire sequence. In both species these features identified IDRs from nucleolar and other RNA-binding proteins and showed statistical association with the GO annotation "ribonucleoprotein complex." In Nop56, the C-terminal segment succeeding the activating region is characterized by lysine repeats (with some acidic amino acids interspersed.) In Nop56, the C-terminal lysine-rich region from E464 to D504 has previously been described as the "K-tail" [91] and is important for interaction with fibrillarin [92]; our mutation map indicates that K454 to D460 is the most important region for Max F244 in the C-terminal IDR of Nop56 (S1 Fig).

To illustrate the power of the global IDR feature space, we explored bulk properties of IDRs associated with subcellular structures/localizations. To do so, we visualized the global enrichment of features in IDRs from clusters of proteins obtained from unsupervised analysis [93] of systematic microscope images of human cells [94]. We first compared IDRs from mitochondrial localized proteins to those from clusters enriched in Golgi and Membrane proteins. As expected, and as found for yeast mitochondrial targeting signals (Fig 3D) the IDRs in human mitochondrial proteins also appear to be rich in hydrophobic and positively charged amino acids (Fig 5E) presumably related to their N-terminal targeting signals. We next explored the IDRs in clusters of proteins enriched for Golgi and membrane localization. Interestingly, we see enrichment of distinct hydrophobic bulk properties (Fig 5E, membrane IDRs are more aromatic) as well as acidic-leucine repeats specific to Golgi and alanine/serine/threonine repeats specific to membrane. We speculate that the later may be the sites of posttranslational modification (as with glycosylation sites in yeast, Fig 4A).

Finally, we compared the features enriched in IDRs from clusters of proteins enriched in the sub-compartments of the nucleus (Fig 5E). We find that the IDRs in each sub-

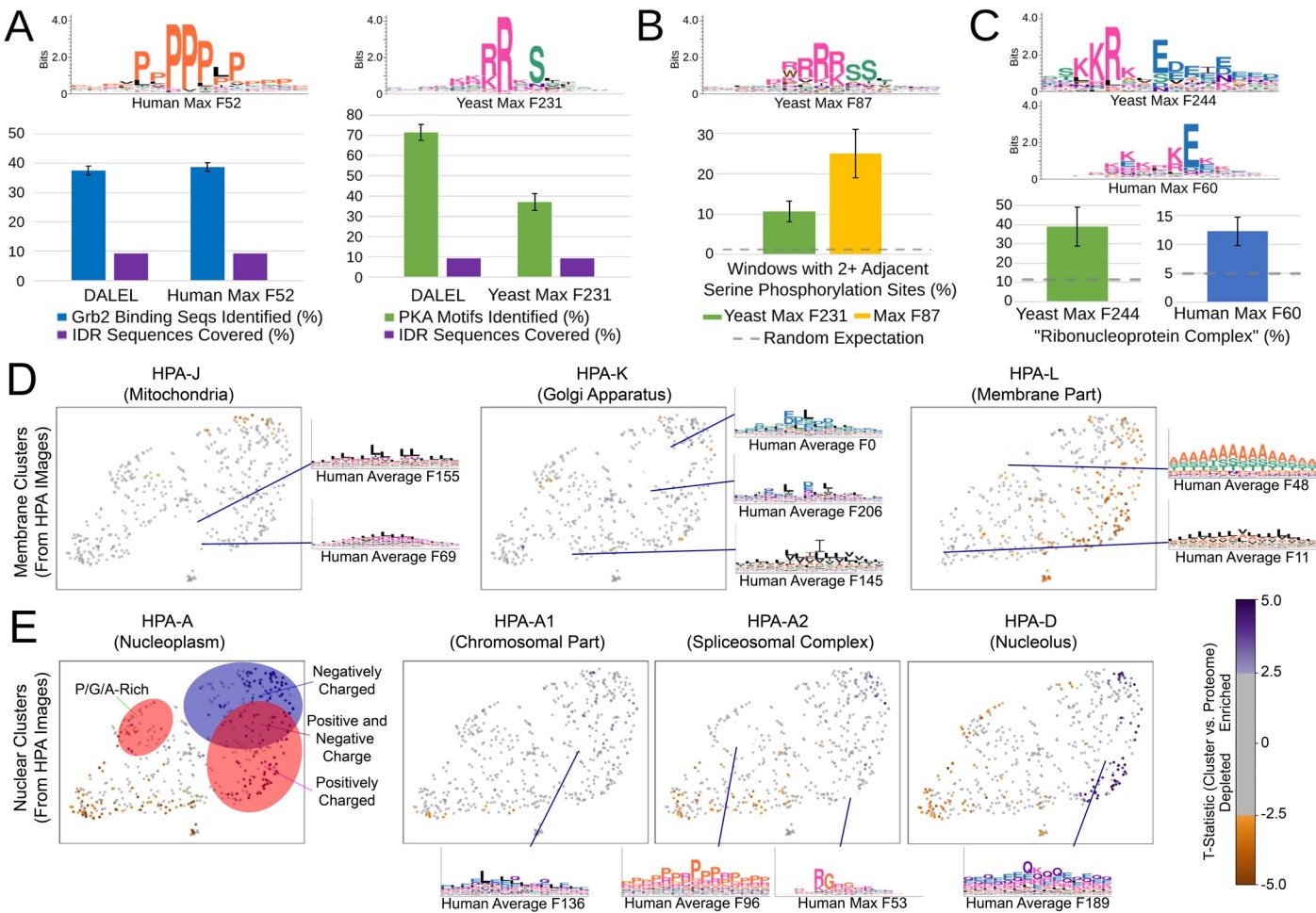

**Fig 5.** (A) Statistical enrichment of reverse homology features points to known motifs for Grb2 and PKA (top left and right, respectively). Bottom: benchmarking reverse homology features against DALEL, a state-of-the-art motif-finder. Recall of residues within characterized binding sites (blue and green bars) at a fixed total number of predictions (purple) is compared. (B) A novel motif (top logo) is more likely to match a peptide with double phosphorylation *in vivo* (gold bar) than random expectation (dashed line) or the feature identified as the cannonical PKA consensus (green bar). (C) Novel "positive to negative charge transition" features (top logos) are more likely to be found in proteins annotated as ribonucleocomplex in both yeast and human models than random expectation (dashed line). In A-C error bars represent standard errors of the proportion using the normal approximation to the binomial. (D and E) Global representations of features enriched in clusters of human proteins obtained through unsupervised analysis of microscopy images (HPA-X). UMAP scatter plots of the feature space are generated as in Fig 2. T-statistics from enrichment of features in the image clusters are indicated by colour and logos show representative examples of enriched features. (D) differences in the bulk properties of IDRs in proteins with different membrane localizations. The enrichments for the mitochondrial IDRs (likely targeting signals) are shown for reference on the left. (E) shows differences between bulk properties of IDRs in various nuclear subcompartments. The enrichments for the nucleus are shown for reference on the left.

compartment are enriched for specific molecular features not enriched in the cluster associated with proteins localized to the nucleoplasm. For example, consistent with the preponderance of RNA binding proteins in splicing and the presence of RG repeats in these proteins [22], we found a max-pool RG feature enriched in the cluster associated with the spliceosome. Interestingly, we also find a proline-arginine repeat, which to our knowledge is not known to be associated with splicing or RNA binding. Similarly, while we find (as expected) highly charged IDRs in the proteins associated with nucleolar localization, we also find a novel glutamine/charged feature enriched in this group. These new molecular features derived from our global analysis illustrate the potential of unsupervised feature discovery from analysis of proteome-scale subcellular localization data.

## Visualization of diverse human IDRs reveals consistency between reverse homology features and known features

Thus far, we focused on discovery and exploration of molecular features in IDRs, which was our primary goal. However, we wondered whether we could visualize the results of the reverse homology approach for individual IDRs. To do so, we obtained the top five most activated features in our reverse homology model (ranked by Z-scores and indicated in the left of Fig 6) for individual IDRs. We first focused on p27 (also known as CDKN1B), because it has a well-studied C-terminal IDR spanning positions 83 to 198 in the sequence, known as the kinase inhibitory domain (p27-KID). This region mediates promiscuous interactions with cyclin-dependent kinase (Cdk)/cyclin complexes through a disorder-to-order transition [95]. We show a summary of known post-transcriptional modification sites and localization signals [96,97] in purple in Fig 6A. We found that the top features for reverse homology (blue trace for average feature, red bars for location of max pool match) appear to overlap with many of these. Furthermore, mutation maps show that important residues for our reverse homology features are consistent with these known sites. For example, Max Feature 71 (Fig 6A.1) overlaps Y88 and Y89, which are modification sites for Src family tyrosine kinases [96]; the letter map indicates that Y89 is the most important residue for this feature. Similarly, Max Feature 241(Fig 6A.3) appears to identify another key phosphorylation site at T157[98]. For comparison, visualizations obtained through ELM [19] reveal a much larger number of residue level predictions, while ANCHOR2[10] (S2 Fig) suggests a large region of predicted binding.

We next considered a C-terminal IDR in hnRNPA1 from positions 183 to 372 as an example of an IDR whose function is thought to be determined by bulk properties as opposed to motifs. This IDR is known as a prion-like domain that facilitates liquid-liquid phase separation [99]. A recent study showed that the uniform patterning of aromatic residues in this IDR is critical to phase-separation, while also inhibiting aggregation [24]. Consistent with these findings, we find that the top three features for our reverse homology model (all average features, two are shown in blue in Fig 6B) are all sensitive to the aromatic amino acids in the sequence of hnRNPA1 (purple bars in Fig 6B). Consistent with this, multiple-sclerosis causing mutations F325L and F333L [100] modify aromatic residues that are highly activating for the top feature, Average Feature 198 (blue trace in Fig 6B). We show the mutation map for Average Feature 32 in Fig 6B.1, which shows that the feature is sensitive to YG or YS repeats in an R/N-rich context, and disfavours negatively charged residues D and E. A recent study analyzed the importance of the context around the aromatic amino acids [101], and found that R can act in a similar way to aromatic residues in stabilizing the interactions in this IDR, while D and E are strongly destabilizing, remarkably consistent with the features we identify. The same study suggests that the glycines and serines are important "spacers" giving flexibility to the amino acid chain between the aromatic residues. Intriguingly, ALS-causing missense variants in hnRNPa1 D314N and N319S [100] are adjacent to F315 and Y318, consistent with the idea that the immediate context of the aromatic residues might be important. However, both of these mutations appear to increase the favourability in this region, perhaps related to enhanced "stickiness" [24] and consistent with increased stress-granule localization observed for disease causing mutants of hnRNPA1[102].

Lastly, we visualized the top features for a more poorly understood, > 400 residue IDR2 from Ataxin-2 [103], between the folded Lsm domains and the conserved PAM2 motif. The most activated feature is a basic max-pool feature following an in vivo phosphorylation site at T624[104]. This also falls within a region that highly activates the second most activated feature, a TP-rich average pool feature. It has been recently suggested that this IDR binds microtubules, and the basic max-pool feature is consistent with the microtubule binding motifs

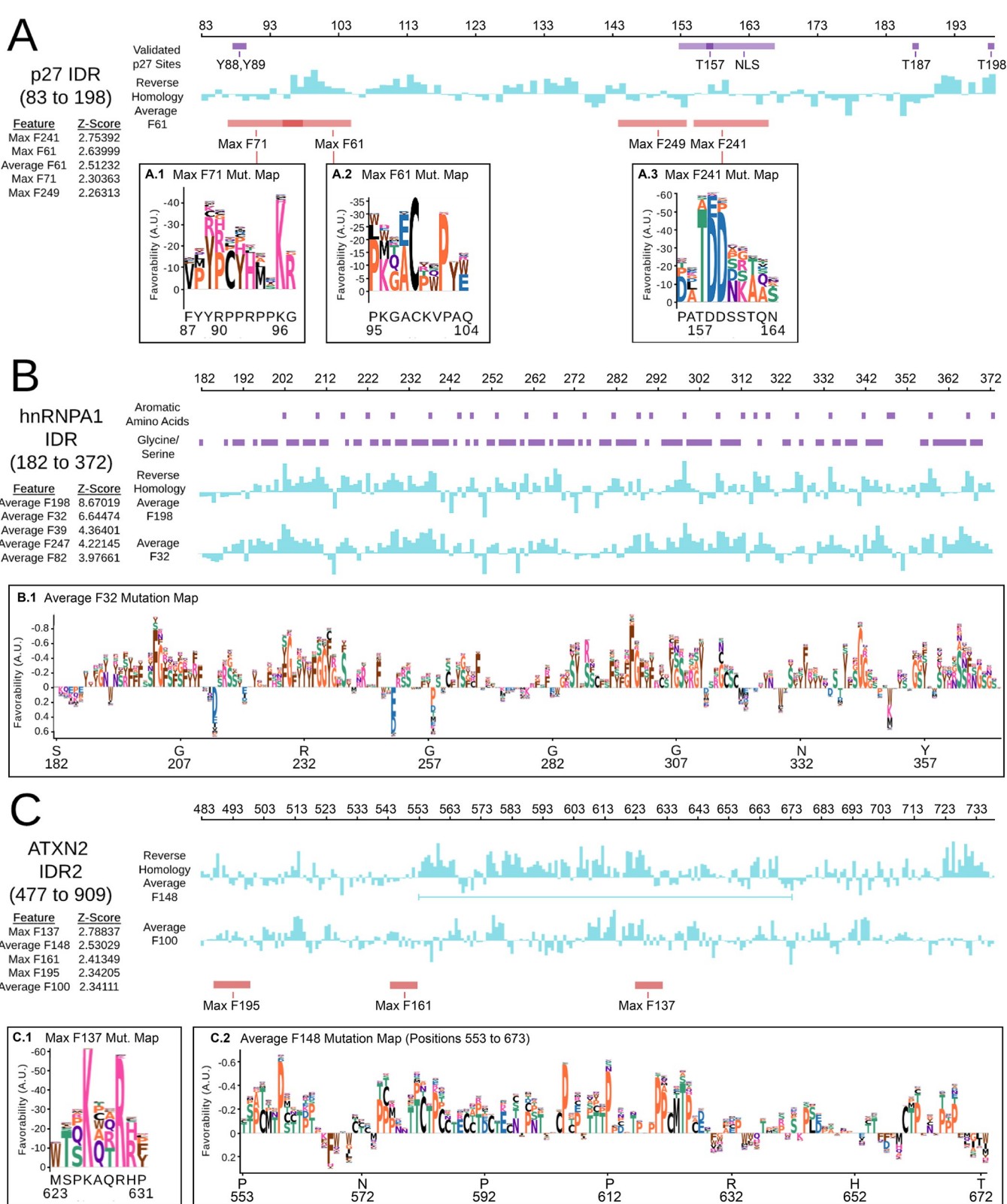

**Fig 6.** Summaries of known features (purple) compared to the top ranked reverse homology features (red and blue) for three individual IDRs, plus letter maps for selected features. We show the position of max pooled features in red (boundaries set using a cut-off of -10 or lower in magnitude), and the values of average features in blue. Average features are sorted in descending order (i.e. the top ranked feature is at the top.) Mutation maps are visualized as in Fig 4.

discovered [103]. On the other hand the importance of threonines and prolines is a new hypothesis, but may be consistent with the abundance of proline-directed phosphorylation in this region [104].

Overall, these cases demonstrate that our reverse homology model learns versatile features and that letter map visualizations of the top features for individual IDRs can yield new hypotheses for both well-characterized IDRs (such as local context around aromatic residues in hnRNPA1), and less characterized IDRs (like prolines in IDR2 from ATXN2).

## Discussion

We present, to our knowledge, the first proteome-wide, evolutionary approach for feature discovery using neural networks. Compared to other systematic homology-based approaches for intrinsically disordered regions [33,34], our method discovers more flexible and expressive features than motifs: we show that our models learn features such as repeats or distributed properties. This expressiveness is important in the context of IDRs, where previous studies have shown that function is often mediated by global "bulk" properties [18]. Like previous comparative genomics methods, our method is systematic, in that it discovers a large set of features informative of many different functions in the proteome, and unbiased by prior knowledge, in that it relies only on automatically-assigned homology to discover features. The latter property sets our method apart from deep learning approaches on protein sequences that use regression problems to train models, such as a recent study that discovered features of disordered activation domains by training deep learning models to predict the results of a transcriptional activation assay [45]. We argue that optimizing models to predict prior knowledge of function or assay measurements that reflect specific aspects of function, will lead to the model learning features for these functions exclusively. In contrast, training a model to predict homology yields a potentially more general set of features that are conserved over evolution.

In many cases, our method learns features that are highly consistent with consensus motifs or bulk features previously known; this congruence is exciting because the model learns independent of prior knowledge, so "re-discovering" this biology supports the claim that our models are learning biological signals. At the same time, even when the model learns features that are consistent with prior expert-defined features, there is often additional subtlety or depth. For example, for our yeast RG-repeat feature Average F65, we showed that the feature has an additional preference for aromatic amino acids, consistent with the recent knowledge that these amino acids mediate phase separation, similar to RG-repeats in these sequences [24,25,87]. Similarly, we showed that our model develops two subclasses of PKA-like consensus motifs, and one is more sensitive to double phosphorylation sites. These examples demonstrate the power of unsupervised analysis to refine previous knowledge.

From a computational biology perspective, we note that many of the individual feature analyses we presented in this study resemble bioinformatics studies that make functional predictions based on conserved motifs or other features [105]. For example, our analysis of PKA phosphorylation sites parallels a previous evolutionary proteomics study that systematically identified PKA substrates [32]: of the 25 of 92 conserved PKA motifs identified that are present in IDRs, our automatically learned yeast PKA feature Max F231 overlaps 10 in the 139 most activated IDRs, suggesting that our feature is in good agreement with this previous study and can also be used to identify putative modification sites. Unlike these studies that start with a known feature and search for new predictions, our approach learns many features in parallel, without having to pre-specify motifs/features of interest. In principle, these bioinformatics analyses can be applied to all of the 512 features learned by our model, enabling hypothesis discovery at an unprecedented scale. (We note that many of the features learned by our model

appear to be redundant with each other (see our annotations in S2 File), so this is an upper bound.) However, like all studies that rely on evolutionary conservation to identify function, we can only identify features that are conserved between homologs: species-specific or lineage-specific functions or features are not expected to be identified through this approach. Development of analysis methods to identify more complex patterns of IDR evolution is an area of increasing research interest [106–108].

Our analysis of individual regions (such as in p27 and hnRNPA1 shown above) indicate that unsupervised deep learning approaches like reverse homology, paired with appropriate interpretation methods, may lead to predictions of functional residues and regions within IDRs. This would be especially useful for IDRs that are mutated in disease, for which we have few mechanistic hypotheses about function [3,109–111]. However, we caution that at this stage, our mutation maps should be thought of as a visualization tool for exploration, as we have not demonstrated their predictive power for individual mutations.

From a technical perspective, reverse homology employs a self-supervised approach, as many emerging representation learning approaches for protein sequences do [50–54]. Unlike these methods, which are mostly based on methods adapted from natural language processing, we proposed a novel proxy task that purposes principles of evolutionary proteomics as a learning signal instead [57]. Another distinction in our study is that previous approaches primarily focus on representation learning, with the aim of optimizing the performance of the representation on downstream regression tasks reflecting protein design or classification problems. In contrast, we focus on feature discovery. We argue that representation learning and feature discovery are distinct aims that require different design philosophies. For example, in this study, we employed a lightweight convolutional architecture, because the interpretation of features is a necessary property. Moreover, we preprocessed the data to remove global information like sequence length or whether the sequence was at the N-terminus: while this information is often useful for downstream tasks, we observed that our models learn fewer "interesting" local features without this preprocessing. In other words, while representation learning does not care about what features are learned as long as they contribute signal to downstream classification or regression problems, we designed our feature discovery approach to learn general, interpretable features that reflect the biology of IDRs.

However, as deep learning architectures are developed for protein sequences, and as new interpretation methods are designed to complement these architectures, updated implementations of our approach with these architectures are also possible. Currently, a major limitation of our convolutional neural network architecture is that it does not capture distal interactions; to some extent, our average pooled features allow for the representation of features distributed across the entire sequence, but these features cannot capture any non-additive interactions. However, transformer architectures are capable of modeling distal interactions, and there has been progress in making these models tailored to multiple sequence alignments, and interpreting the self-attention modules [112,113]. A second limitation is that our convolutional model requires sequences to be standardized in length as input. This preprocessing requirement means that we may lose key elements of longer sequences. Shorter sequences require padding; we used "repeat" padding, since we found with a special padding token the neural network can use cues about length to trivially eliminate many possible proteins from the contrastive task, but this runs the risk of creating new spurious repeats. Recurrent architectures address this limitation and allow for arbitrary-sized inputs [50,52]. Overall, integrating these kinds of advances with our method in future work is expected to make the model more expressive, increasing the scope of the features we can discover. A more general limitation to feature discovery using neural networks is that retraining models from random initializations often lead to different features being identified. For example, for some initializations and/or data filtering

heuristics we found features that were highly activated by the phosphorylation site at T198 in p27, while in the model presented above we have a feature activated by T157; for some models Y88 was most activating, while others Y89. We believe this is because the neural network will only learn the minimum feature set needed to solve the reverse homology task; it does not need to learn all the features needed for biological function. Further research is needed to address how to encourage the neural network to consistently learn more of the important features, perhaps by adaptively increasing the difficulty of the contrastive task, or by including multiple training tasks as in pre-training for natural language models, e.g., adding masked-token prediction [114,115]. Finally, further work on interpretation methods would also improve our ability to extract insights from these models: currently, a limitation of our sequence logos for average features is that features that favor specific combinations of repeats (e.g., RG-repeats) are difficult to distinguish between features that favor amino acid content (e.g., R and G content). Though there is clearly more work to be done, we believe the expressive capacity of neural networks combined with biologically motivated training paradigms will enable global unsupervised discovery of functional features in poorly understood biological sequences.

## Methods

### Details of reverse homology

In this paper, we studied sets of automatically obtained homologous IDRs. We concentrate on IDRs in this work, but our contrastive learning task can easily be extended to other definitions of homologous sequences including full protein sequences or structured domains. We will use these sets of homologous sequences as the basis of our self-supervised task.

Let $H_i = \{s_{i,1}, \ldots, s_{i,n}\}$ be a set of homologous sequences. We define a set of query sequences, $S_q$ such that all sequences in the query set are homologous to each other, so $S_q \subset H_i$. Then, we define a set of target sequences associated with the query set, $S_t = \{s_{t-,1}, \ldots, s_{t-,m-1}\} \cup \{s_{t+}\}$ where $s_{t+}$ is a held-out homologue $s_{t+} \in H_i$, $s_{t+} \notin S_q$ and $s_{t-}$ are not homologous to the query set, $s_{t+} \in H_j$, $j \neq i$.

Let $g_1$ be a function that embeds $S_q$ into a latent feature representation, so $g_1(S_q) = z_1$, and $g_2$ be a function that embeds members of $S_t$ into a latent feature representation, so $g_2(s_t) = z_2$ (in this work, $g_1$ and $g_2$ are convolutional neural network encoders.) Our task is to optimize the categorical cross-entropy loss, also commonly called the InfoNCE loss in contrastive learning literature [59], where $f(g_1(S_q), g_2(s_t))$ is a score function (in this work, we use the dot product):

$$\mathcal{L}_{NCE} = -\mathbb{E}_{S_q, s_{t+}, s_{t-}} \left[ log \frac{\exp(f(g_1(S_q), g_2(s_{t+})))}{\exp(f(g_1(S_q), g_2(s_{t+}))) + \sum_{j=1}^{M-1} \exp(f(g_1(S_q), g_2(s_{t-j})))} \right]$$

### Implementation of reverse homology

In principle, the sequence encoders $g_1$ and $g_2$ are flexibly defined, and state-of-the-art neural network architectures are possible, e.g., transformer or LSTMs [50–52]. However, since a priority of this work is interpreting the features learned by our model (see below for details on interpretation approaches), not necessarily to learn the most useful or complete representation possible, we decided to implement our encoders as low-parameter convolutional neural networks (CNNs), a relatively simple architecture that was relatively fast to compute.

Both encoders $g_1$ and $g_2$ begin with three convolutional layers; to capture motifs and residue composition effects, the first layer contains neurons with kernel sizes 1, 3, and 5 [116]. After the convolutional layers, we max and average pool convolutional features over the length of the entire sequence, to capture motif-like and bulk features as discussed above. The max and

average-pooled features are concatenated and fed into two fully-connected layers with output dimension 256, to allow arbitrary combinations of them before prediction. To ensure the model can use both feature types, we scale the average-pooled features by a factor of the post-processed input sequence length divided by the receptive field of the final convolutional layer (17.06 times in this specific architecture) to put the average and max pooled features on the same numerical scale. The output of the final fully connected layer is considered the feature representation. We average the feature representation for all homologues in the query set $S_q$, and calculate the dot product between this average and the representation for each sequence in the target set $s_t$. This dot product is considered our score function $f(g_1(S_q), g_2(s_t))$ in the InfoNCE loss (i.e. the largest dot product is considered the model's prediction of which sequence in the target set is homologous to the sequences in the query set.) The total number of parameters is 1,479,384.

For our implementation, we use a query set size of 8, and a target set size of 400. Using 8 sequences, we expect that features specific to any one homolog and not shared across all homologues will be averaged out. In theory, increasing the size of the target set tightens the lower bound on maximizing mutual information [59].

## Training datasets

To train our model, we used sets of homologous yeast IDRs previously defined by Zarin *et al*. [18]. Briefly, this dataset was produced by aligning orthologues (one-to-one homologues from different species) of yeast proteins previously calculated by the Yeast Gene Order Browser [117]. DISOPRED3 was used to annotate IDRs in *Saccharomyces cerevisiae* sequences, and alignments are used to identify the boundaries of the IDRs across species, (but not supplied to the models during training, i.e. the input to the neural network is the ungapped IDR sequence). Residues in other species that fell within these regions were considered homologous after some quality control steps (see [18] for details.) We filtered this dataset by removing sequences under 5 amino acids, with undetermined amino acids ("X") and/or non-standard amino acids, and only kept homolog sets with more than 9 sequences represented. Overall, this dataset consists of a total of 94,106 IDR sequences distributed across 5,306 sets of homologues.

In addition to our yeast model, we trained another reverse homology model on UniProt reference human protein sequences (downloaded on September 2019 [104]) using SPOT-Disorder v1 [118] for disorder prediction. To ensure that the sequences did not contain any structured domains, we filtered the sequences based on matches in Prosite [119]. We removed human IDRs that had a match in Prosite longer than 10 amino acids. We performed an all by all blastp [14] search to confirm that the human IDRs were not dominated by recent gene duplicates or highly conserved portions of IDRs which would make the reverse homology proxy task trivial. As expected, we found only 18% (2925/15996) of these sequences showed detectible homology (E-value $<10^{-5}$) to another IDR within the set. Homologous proteins were obtained from the OMA homology database [48]. These sequences were aligned using MAFFT [120], and the disorder boundaries of the human sequence were used as a reference to annotate putative disordered regions in the set of homologs by clipping out the region of the sequence that aligned to the human IDR. Because the number of homologs for each human protein varies greatly in the OMA, we used the alignment of each IDR to the human reference IDR to apply a series of heuristic filtering steps to improve the uniformity of the dataset to make it more similar to our previously published yeast data. First, we removed any IDR homologs with 'X' characters or that were 3x longer or shorter than the human reference. For the remaining homologs, we computed pairwise evolutionary distances to the human reference

IDR using a method of moments estimator [121] for Felsenstein's 1981 distance [122], adapted to amino acids with frequency parameters set to the amino acid frequencies of human IDRs. We then use the following heuristic to select sequences. Starting with the closest homolog, we add homologs iteratively according to their distance to the reference, excluding any homologs where the distance from the closest homolog included so far is less than 5x the distance from the human reference IDR. This removes "redundant" sequences that are close to each other, but does so in a way that scales with the distance to the reference. We stop adding homologs when the total (pairwise) evolutionary distance reaches at least 30 substitutions per site, or we run out of homologs. This total evolutionary distance was chosen to be similar to the previously published yeast dataset described above. This dataset consists of 634,406 sequences distributed across the 15996 sets of homologues, and has been made available on the zenodo site supporting this work.

Finally, we trained a reverse homology model on experimentally verified IDRs, by using sequences from the Disprot [7] database (2022_03 release). We retrieved homologs for proteins in Disprot from Ensembl [47], requiring at least one homolog, which we then aligned with MAFFT [120] as above. The disorder boundaries corresponding to the sequence in Disprot were used as a reference to annotate disordered regions for its homologs, by clipping out the region of the sequence that aligned to the Disprot IDR. To filter out repeated entries for IDRs in Disprot, we removed any IDRs that overlapped any amino acids, keeping only the longest IDR. Finally, we applied the same heuristic filters for redundancy between homologs as we did for the human sequences. These operations resulted in a total of 53,050 sequences distributed across 1,467 homologs, which are also available on the Zenodo archive supporting this work.

## Preprocessing and training

We one-hot encoded sequences as input into our models. To standardize the lengths of the sequences, if the sequence was longer than 256 amino acids, we used the first and last 128 amino acids from the sequence. If the sequence was shorter, we "repeat padded" the sequence until it was over 256 amino acids (e.g. in this operation "ACD" becomes "ACDACD" after the first repetition), and clipped off excess length at the end of the padded sequence. To test the effects of using only the first and last 128 residues, we also trained a model using the central 256 amino acids and found that the majority of the features were strongly correlated between the two models (S3 Fig). IDRs >256 residues comprise a small fraction of the dataset (6.2% for yeast and 13% for human).

In our preprocessing operations, we sought to reduce the impact of certain global properties, which while potentially biologically informative, would allow the model to rule out the majority of non-homologous sequences in the target set on the basis of relatively trivial features for most query sets, reducing the effectiveness of our contrastive task. One is the length of the sequence, motivating our use of the repeat padding operation, which reduces cues about length compared to the use of a special padding token. The other global feature we identified was whether the IDR occurs at the start of a protein or not, as indicated by a methionine (from the start codon) at the beginning of the IDR. We clipped this methionine from the sequence if the IDR was at the N-terminus of a protein.

We trained models for 1,000 epochs using the Adam optimizer [123], where each epoch iterates over all sets of homologues in the training dataset. For each set of homologues, we randomly drew 8 unique sequences at each epoch to form the query set and 1 non-overlapping sequence for the target set. To save memory and speed up training, we used a shared target set for each batch of query sets: homologous sequences for one query set in the batch would be

considered as non-homologous sequences for the other query sets. If the target set size is larger than the batch size (as it was in our experiments), the remainder of non-homologous sequences are sampled at random from homolog sets not used in the batch. We trained models with a batch size of 64, and a learning rate of 1e-4.

Classification of homologs is a proxy-task in our setting and predictive power in this task is not necessarily correlated with the biological relevance of the features derived. Nevertheless, we performed a simple "held out" validation benchmark to confirm that the model was learning generalizable features (S4 Fig). We did a 90/10 split for training/validation and then, to ensure that the estimates of power in these analysis was not biased by sequence similarity, we removed from the validation set any homolog sets where the reference (yeast or human) IDR showed detectible homology with any reference IDR sequence in the training set. We found that for both human and yeast models, the classification on held out homologous IDR sets was relatively high (50% and 70% for yeast and human models respectively; note that the expectation for a random classifier is 1/400 on this task).

## Correlation with literature-curated features

To compare our features against literature-curated features, we binarized all amino acid positions in all of the IDR sequences in our yeast data using each of these regular expressions. Amino acids that are contained in a match to the regular expression are assigned a value of 1, while all other amino acids are assigned a value of 0. We calculated the global correlation between these binarized positions and the activation value of neurons in our convolutional neural network at each position. A higher correlation indicates that a neuron outputs high feature values at positions that match the regular expression of a literature-correlated feature and low values at positions that do not match.

## Interpretation

To interpret the features learned by our model, we adapted two previous interpretation methods. First, we use the approach proposed by Koo and Eddy for generating motif-like visualizations by collecting the parts of sequences that maximally activate neurons to calculate position frequency matrices [44]. We collect sequences that reach at least 70% of the maximum activation for that neuron. If there are less than 20 sequences meeting this threshold, we instead collect the 20 highest activating sequences. For max-pooled features, we collect the maximally activating subsequence, and add all amino acids in this window to the PFM with equal weight. For average-pooled features, we add all windows to the PFM, but weigh all windows in the sequence by the activation for that window divided by the activation of the maximally activating window in that sequence. These PFMs are converted to a position probability matrix and visualized as sequence logos using the Biopython package, modified with a custom color scheme [124]. Unlike Koo and Eddy [44], we do not discard windows that overlap with the start or end of a sequence, to avoid too few inputs due to our larger receptive field and smaller sequence sizes; we simply do not let parts of the sequence overlapping the start and end of the sequence contribute any frequency to their corresponding position in the position frequency matrix. Overall, this method produces a sequence logo for each neuron, summarizing the kinds of subsequences that activate the neuron. This method can only be applied to convolutional layers before pooling, because it requires us to measure the activation at specific positions in a sequence; in our experiments, we apply it to the final convolutional layer in our models (Conv3 in Fig 3).

We also adapted mutation maps, were we computationally introduce every possible point mutation in a sequence [39]. For a given feature and IDR, we substitute each amino acid in the

IDR sequence to each other amino acid and measure the change in the value of the feature. The runtime to create a mutation map for ATXN2 is 209 seconds on an Nvidia Quadro P6000. We visualize the mutation map as a sequence logo, which we term a "letter map" to distinguish them from the standard sequence logos that show information content in multiple sequence alignments [83]. In this letter map, any amino acids that would generally reduce the value of the feature if mutated is shown above the axis, while any amino acids that would generally increase the value of the feature if mutated are shown below the axis. The combined height of the letters corresponds to the overall magnitude of the increase or decrease mutating the position would induce on the feature. For positions above the axis, we show the amino acids that are most permissible in that position. For positions below the axis, we show the amino acids that are least permissible in that position. In summary, letters above the axis are favored residues in favored positions, while letters below the axis are disfavored residues in disfavored positions. We used the Logomaker package, modified with a custom color scheme, to visualize these letter maps [125]. More details and formulas for these letter maps are available in S1 Methods.

We also reported the enrichment of features for GO enrichments [77] and we note that these are done using a background of all proteins with IDRs (not all proteins), to avoid spurious enrichments.

## Comparison with a state-of-the-art motif-finder

Following the benchmarking experiments on Grb2 binding sites [88] we use the number of residues identified within characterized sites as the "recall" at a fixed "coverage" or total number of residues predicted. We used the web implementation of DALEL [88] to identify motifs in the IDRs only. To identify the "top" reverse homology feature, we simply performed a t-test for the activation of each feature comparing the IDR set (either Grb2 or PKA targets) to the proteome. To obtain residue-level predictions, we smoothed the activations of the top feature using a Gaussian Kernel with width 15 residues, and ranked the residues by this smoothed activation. As with DALEL, we took the number of residues within characterized sites as the "recall" when we used a fixed "coverage", chosen to match DALEL.

## Code and data availability

Code for training our models and visualizing/extracting features is available under a CC-BY license at github.com/alexxijielu/reverse_homology/. Pretrained weights for our models, fasta files of IDR sequences used to train both models, and labels for IDRs used in our classification benchmarks are available at zenodo.org/record/5146063.

## Supporting information

**S1 Fig.** The C-terminus of yeast Nop56 contains alternating regions of positive and negative charge that activate yeast Max-pool Feature 244 A) The sequence logo for Max Feature 244 (Max F244) B) The mutation map (as in Fig 4) showing the importance of residues for activation of this feature in a section of the Nop56 C-terminal IDR.
(PNG)

**S2 Fig.** Predictions for p27 for ANCHOR2 (A) and ELM (B). For both predictions, we inputted the full protein, so we highlight the C-terminal IDR in gold. A) The blue line shows the ANCHOR2 score predicting disordered binding regions. B) The blue boxes show matches to short linear motifs within the sequence, as labeled on the left. The darker the blue, the more conserved the motif is across orthologues. The red circles indicate known instances annotated from literature.
(PNG)

**S3 Fig. Effects of IDR sequence cropping on features learned by reverse homology.** The distribution of the maximum correlation of features learned using the cropping heuristic used in the main text (128 residues at the start/end of sequence) and features learned in a model using an alternative cropping heuristic (256 in the center of sequence). Y-axis shows the number of features with that maximum correlation. Most features have a feature with at least correlation of 0.7. See Methods for more details.
(PNG)

**S4 Fig. Training and validation accuracy for the reverse homology proxy task.** A) Model trained on yeast IDR homolog sets. B) Model trained on human homolog sets. See Methods for more details C) Model trained on experimentally characterized IDRs from DisProt. In this case, the validation set was 1000 randomly selected human homolog sets from the SPOT-Disorder human IDR predictions.
(PNG)

**S1 File. Top activating subsequence or IDR from max- and average-pooled features respectively.**
(XLSX)

**S2 File. Compressed archive including HTML table of all feature logos for yeast features and feature representation for each yeast IDR.**
(TAR.GZ)

**S3 File. Known yeast features and correlations with features from the reverse homology model trained on yeast IDRs.**
(XLSX)

**S4 File.** Details of datasets used for benchmarking and Tables A to C, which include all benchmarking results. Table A in S4 File: Performance of representations on classifying mitochondrial targeting signal IDRs and IDRs containing Cdc28 phosphorylation sites, with nearest-neighbor and logistic regression classifiers. We report standard error. Table B in S4 File: Average performance of representations in classifying 23 previous computationally annotated clusters by Zarin et al., with nearest-neighbor and logistic regression classifiers. We report standard error. Table C in S4 File: Median fold enrichment of representations for GO Slim annotations in nearest neighbor classifiers, using 92 GO Slim annotations.
(DOCX)

**S5 File. Compressed archive including HTML table of all feature logos for human features and feature representation for each human IDR.**
(TAR.GZ)

**S6 File. Known yeast features and correlations with features from the reverse homology model trained on human IDRs.**
(XLSX)

**S7 File. Compressed archive including HTML table of all feature logos for features trained on DisProt IDRs.**
(ZIP)

**S8 File. Known yeast features and correlations with features from the reverse homology model trained on DisProt IDRs.**
(XLSX)

**S1 Methods. Details of the theory behind reverse homology and calculation of the letter maps.**
(DOCX)

## Acknowledgments

We thank Caroline Uhler, Karren Dai Yang, and Kevin Yang for valuable feedback.

## Author Contributions

**Conceptualization:** Alex X. Lu, Amy X. Lu, Alan M. Moses.

**Data curation:** Alex X. Lu, Iva Pritišanac.

**Formal analysis:** Amy X. Lu.

**Funding acquisition:** Alex X. Lu, Alan M. Moses.

**Investigation:** Alex X. Lu, Amy X. Lu, Alan M. Moses.

**Methodology:** Alex X. Lu.

**Project administration:** Julie D. Forman-Kay, Alan M. Moses.

**Resources:** Iva Pritišanac, Taraneh Zarin, Alan M. Moses.

**Software:** Alex X. Lu, Iva Pritišanac.

**Supervision:** Julie D. Forman-Kay.

**Validation:** Alex X. Lu.

**Visualization:** Alex X. Lu.

**Writing – original draft:** Alex X. Lu.

**Writing – review & editing:** Alex X. Lu, Iva Pritišanac, Taraneh Zarin, Julie D. Forman-Kay, Alan M. Moses.

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
