## [Decision Letter · Decision Letter 0]

3 Nov 2021

Dear Dr. Moses,

Thank you very much for submitting your manuscript "Discovering molecular features of intrinsically disordered regions by using evolution for contrastive learning" for consideration at PLOS Computational Biology.

As with all papers reviewed by the journal, your manuscript was reviewed by members of the editorial board and by several independent reviewers. In light of the reviews (below this email), we would like to invite the resubmission of a significantly-revised version that takes into account the reviewers' comments.

We cannot make any decision about publication until we have seen the revised manuscript and your response to the reviewers' comments. Your revised manuscript is also likely to be sent to reviewers for further evaluation.

Sincerely,

Damiano Piovesan

Guest Editor

PLOS Computational Biology

Arne Elofsson

Deputy Editor

PLOS Computational Biology

Reviewer's Responses to Questions

**Comments to the Authors:**

Reviewer #1: The authors present a neural network based feature discovery method for IDRs, which they called "reverse homology" method. The model aims to exploits the evolutionary principle that functional features can be conserved in IDRs. Although the approach is very interesting, the presentation of the method is not clear and the results are not convincing enough.

Major: 

Importantly, more details are needed to understand better how the reverse homology method works. For instance, it is unclear what the output of the neural network is. Also, more details about the measurement of the effect of mutations in the in-silico mutational scanning are necessary. Although the main purpose the method is not the detection of homologous sequences directly, it would be important the test the method's performance for the original task through cross-validation.

The example for p27 nicely presents the capability of the method to retrieve functionally relevant features for human IDRs. However, the other examples, especially the identification of cysteine repeats in hair keratin-associated proteins are less convincing. Presentation of other interesting human examples, like p27, could increase the biological value of this work.

Minor:

Small typos can be found in the manuscript:

Figure 1. legend: "D) This panel..."

Figure 6. legend: "For the heat maps..." (a bracket is absent)

Figure 6B.: Nop1 is indicated instead of Nop65

Position boundaries of Nop56 are not correct ("...K464 to D460 is the most important region...")

Reviewer #2: Review uploaded as an attachment

Reviewer #3: The manuscript by Lu et al. describes a novel method to learn conserved sequence features in groups of homologous intrinsically disordered regions (IDRs) using unsupervised machine learning. This is potentially of interest, as more methods for the functional characterization and classification of IDRs are needed. The manuscript is unfortunately written in a style which tries to emphasize novelty by using a set of non-standard jargon terms and using many references to previous work which obfuscate rather than clarify its content. The text is also lengthy and dense, dwelling on relatively minor aspects while glossing over some key points which would help to evaluate its technical quality. It raises a number of major concerns:

* Introduction. This presents a rather narrow view of the field, ignoring several key computational resources for IDRs and giving the impression that much work is still to do when it is really not. Specifically, the authors fail to mention and major database for IDRs, e.g. DisProt or MobiDB. These are important as a gold standard for IDRs (DisProt) and to facilitate large-scale analysis as well as functional annotation for IDR "flavors" (MobiDB). Instead, the authors largely limit their description of computational work to some self-references. This has to be corrected in order to appreciate the true novelty and impact of their contribution. Lastly, since the authors are proposing a neural network for IDRs, the recent CAID experiment is of relevance (Necci et al., Nature Methods 2021).

* Methods.

1. Terminology. While the use of "reverse homology" may overstate the novelty but can be justified, several other "inventions" do not: "letter map" has been called sequence logo or enrichment plot by the authors of the software used (LogoMaker); "in silico mutational scanning" really means a sequence permutation. These jargon terms should be replaced by standard ones to make the paper easier to understand.

2. Implementation of reverse homology. This section is awkward to read. There is both a lengthy description of previous work, which does not belong in a materials and methods section, while several crucial elements are missing. If this reviewer understood correctly, the authors are proposing a plan convolutional neural network (CNN) trained to recognize IDRs. The details on this CNN should be spelled out clearly, e.g. using the recent DOME recommendations (Walsh et al., Nature Methods 2021). Last but not least, the sequence encoding hinted at in Figure 1D is not explained. Without further details it is impossible to evaluate the technical merit of this implementation.

3. Training datasets. This section contains too many references to previous work, which hamper understanding and may obfuscate crucial details. E.g. what was the pairwise sequence identity within and between IDR groups? How do the authors treat "jumpy" alignments, with either lots of gaps or widely varying predicted IDRs across homologs? Is the presence of a bona fide IDR across homologs even checked? The choice of two different IDR predictors for the two datasets also needs an explanation (even though CAID, see above, may help).

4. Preprocessing and training. The effect of cutting up the IDR to the first and last 128 residues has to be investigated. While the reasoning to avoid length differences is clear, the knock-on effects caused by this are not. This has to be spelled out clearly. What is being learned? Is the CNN "just another" IDR predictor for which single internal neurons are interpreted? This would be both novel (interpretation) and not so novel (predictor) at the same time.

5. Correlation with literature-curated features. How do the authors derive these? from single publications and only for the test cases described in the text? A more systematic approach is needed in order to have meaningful benchmarking, e.g. by comparing with DisProt annotations. This limitation makes Figure 3 unnecessarily difficult to evaluate and understand.

6. Benchmarking and comparison to other methods. This is completely glossed over. Why is there no comparison to other methods? This is easy enough to arrange and at the very least some baseline approaches are needed. E.g. how well does the method perform compared to BLAST or even simple pairwise sequence alignment? How does plain sequence conservation work on the IDRs? How well do motif-based methods work, e.g. ELM or Dilimot? The latter is for de novo motif identification, which is clearly what the authors are doing as well. It is surprising to see that no attempt was made to benchmark the method.

7. Sequence features identified. The authors claim that their method learns "functional features", with "motifs" as well as "bulk properties". However, from the figures it is only possible to discern what look remarkably like sequence motifs, albeit not regular expressions.

8. In silico mutational studies. If this reviewer understood correctly, this somewhat hyperbolically named analysis presents different amino acid compibantions to the CNN and records the drop in scores. This is useful but should be described for what it is. It is NOT an experimental validation and can, in principle, be carried out on any neural network-based IDR predictor. The case for its usefulness should therefore be more clearly argued and/or the statements regarding it toned down (the latter applies widely throughout the text).

9. Biological examples. These are probably too much of a good thing and distract from the method. Rather than using a few case studies to argue in detail the nuances of their predictor, the authors should focus on validating the method at large. In light of this, the enthusiastic tone of the text should also be toned down.

10. Preprints. Several key statements in the manuscript are backed up by citations of preprints, including several self-citations. While this can sporadically be accepted, and indeed contribute to the "timeliness" of the paper, in its current form it is beyond the pale. This reviewer has counted a dozen preprints in the reference section. Checking these, in combination with the previously mentioned tendency to obfuscate the meaning with jargon and lack of key details, places an undue burden on unpaid reviewers trying to districate the true contribution of this manuscript.

11. Discussion. Statements referring to "excitement" and "urgent needs" need to be toned down in accordance with the previous remarks. Also, the authors have built a predictor and not made a "comparative proteomics" study.

Specific points

a. Introduction. Tools and databases are not cited, e.g. BLAST and Pfam.

b. Sequence "homology". This should be referred to as either "homology" or "sequence similaritty/identity". Homology is not a quantitative term.

**Have the authors made all data and (if applicable) computational code underlying the findings in their manuscript fully available?**

Reviewer #1: None

Reviewer #2: Yes

Reviewer #3: **No: **The authors state that "all data will be made available at the author website".

PLOS authors have the option to publish the peer review history of their article (what does this mean?). If published, this will include your full peer review and any attached files.

Reviewer #1: No

Reviewer #2: No

Reviewer #3: No
---

## [Decision Letter · Decision Letter 1]

12 Apr 2022

Dear Dr. Moses,

Thank you very much for submitting your manuscript "Discovering molecular features of intrinsically disordered regions by using evolution for contrastive learning" for consideration at PLOS Computational Biology. As with all papers reviewed by the journal, your manuscript was reviewed by members of the editorial board and by several independent reviewers. The reviewers appreciated the attention to an important topic. Based on the reviews, we are likely to accept this manuscript for publication, providing that you modify the manuscript according to the review recommendations.

All reviewers were positive about the revised version of the manuscript, I agree with them and support the publication of the work. Please note the comments of Reviewer 1 that might be useful for the proof.

Sincerely,

Arne Elofsson

Deputy Editor

PLOS Computational Biology

Arne Elofsson

Deputy Editor

PLOS Computational Biology

[LINK]

All reviewers were positive about the revised version of the manuscript, I agree with them and support the publication of the work. Please note the comments of Reviewer 1 that might be useful for the proof.

Reviewer's Responses to Questions

**Comments to the Authors:**

Reviewer #1: The Authors have done significant improvements in the revised manuscript that have made the study more convincing. They have revised the explanation of how their neural network based method works, which is one of the most important parts of the work. Taken together, the improved version is clear, the results are easier to interpret now.

Minor points:

• The method should be also tested on experimentally verified disordered regions (e.g. Disprot database)

• Small typos:

“Overall, we find that while some GO Slim categories are highly enriched with both feature sets

(e.g. “Cell wall organization”, “Translation” or “Ribosome biogenesis”, highlighted in Figure

3A)…” - figure reference is not correct.

“Examination of the of the most activating”

Reviewer #2: The authors have done an excellent job addressing the reviewer comments (some of which I felt were somewhat counterproductive). I have no further comments and strongly support publication.

Reviewer #3: The authors have significantly improved the manuscript after revision. All concerns have been answered extensively and satisfactorily. Good job.

**Have the authors made all data and (if applicable) computational code underlying the findings in their manuscript fully available?**

Reviewer #1: None

Reviewer #2: Yes

Reviewer #3: Yes

PLOS authors have the option to publish the peer review history of their article (what does this mean?). If published, this will include your full peer review and any attached files.

Reviewer #1: No

Reviewer #2: No

Reviewer #3: No

Figure Files:

Data Requirements:

Reproducibility:

References:

---

## [Editor Report · Decision Letter 2]

23 May 2022

Dear Dr. Moses,

We are pleased to inform you that your manuscript 'Discovering molecular features of intrinsically disordered regions by using evolution for contrastive learning' has been provisionally accepted for publication in PLOS Computational Biology.

Best regards,

Damiano Piovesan

Guest Editor

PLOS Computational Biology

Arne Elofsson

Deputy Editor

PLOS Computational Biology

---

## [Editor Report · Acceptance letter]

23 Jun 2022

PCOMPBIOL-D-21-01432R2 

Discovering molecular features of intrinsically disordered regions by using evolution for contrastive learning

Dear Dr Moses,

I am pleased to inform you that your manuscript has been formally accepted for publication in PLOS Computational Biology. Your manuscript is now with our production department and you will be notified of the publication date in due course.

With kind regards,

Agnes Pap
